# A Comparative Study of Microbial Communities, Biogenic Amines, and Volatile Profiles in the Brewing Process of Rice Wines with Hongqu and Xiaoqu as Fermentation Starters

**DOI:** 10.3390/foods13152452

**Published:** 2024-08-02

**Authors:** Yingyin Yan, Zihua Liang, Yujia Huo, Qi Wu, Li Ni, Xucong Lv

**Affiliations:** 1Food Nutrition and Health Research Center, School of Advanced Manufacturing, Fuzhou University, Jinjiang 362200, China; 218527276@fzu.edu.cn (Y.Y.); 220820089@fzu.edu.cn (Z.L.); 230820094@fzu.edu.cn (Y.H.); 200827089@fzu.edu.cn (Q.W.); nili@fzu.edu.cn (L.N.); 2Institute of Food Science and Technology, College of Biological Science and Engineering, Fuzhou University, Fuzhou 350108, China

**Keywords:** rice wine, fermentation starter, microbial community, biogenic amines, higher alcohols, volatile flavor components

## Abstract

Rice wine is primarily crafted from grains through saccharification and liquification with the help of Qu. Qu plays an important role in the formation of the flavor quality of rice wine. Hongqu and Xiaoqu represent two prevalent varieties of Qu that are typically utilized in the brewing process of rice wine and play a crucial role in its production. In this study, GC, GC-MS, HPLC, and metagenomic sequencing techniques were used to contrast the microbial flora, biogenic amines, and aroma characteristics developed during the fermentation of rice wines, with Hongqu and Xiaoqu being used as initiating agents for the brewing process. The results show that the content of higher alcohols (including n-propanol, isobutanol, 3-methyl-1-butanol, and phenethyl alcohol) in rice wine brewed with Xiaoqu (XQW) was significantly higher than that in rice wine brewed with Hongqu (HQW). Contrarily, the concentration of biogenic amines in HQW surpassed that of XQW by a notable margin, but tyramine was significantly enriched in XQW and not detected in HQW. In addition, a multivariate statistical analysis revealed distinct disparities in the constitution of volatile components between HQW and XQW. Hexanoic acid, ethyl acetate, isoamyl acetate, ethyl caproate, ethyl decanoate, 2-methoxy-4-vinylphenol, etc., were identified as the characteristic aroma-active compounds in HQW and XQW. A microbiome analysis based on metagenomic sequencing showed that HQW and XQW had different dominant microorganisms in the brewing process. *Burkholderia*, *Klebsiella*, *Leuconostoc*, *Monascus,* and *Aspergillus* were identified as the primary microbial genera in the HQW fermentation period, while *Pediococcus*, *Enterobacter*, *Rhizopus*, *Ascoidea,* and *Wickerhamomyces* were the main microbial genera in the XQW brewing process. A bioinformatics analysis revealed that the concentrations of microbial genes involved in biogenic amines and esters biosynthesis were significantly higher in HQW than those in XQW, while the content of genes relevant to glycolysis, higher alcohol biosynthesis, and fatty acid metabolism was significantly higher in XQW than in HQW, which are the possible reasons for the difference in flavor quality between the two kinds of rice wine from the perspective of microbial functional genes.

## 1. Introduction

As an exemplary product in the field of traditional Chinese fermented foods in China, rice wine (also known as *Huangjiu*) is a unique alcoholic beverage utilizing grain as its cornerstone ingredient. It undergoes the parallel process of saccharification and liquification and uses Qu as a starter to promote fermentation [1,2]. Rice wine is popular in China for its characteristic flavor, abundant nutrition, low alcoholicity, and health-promoting properties [3]. Qu plays a crucial role in the formation of the flavor quality of rice wine [4]. Based on the variety of Qu utilized, Chinese rice wine can be categorized into three distinct types, Hongqu rice wine, Xiaoqu rice wine, and Wheatqu rice wine, each of which has its own characteristics [5]. Wheat Qu is made from raw wheat, which naturally contains mold, bacteria, and yeast. In the production process, the wheat is usually crushed, mixed with water, and pressed into brick-like blocks by manual trampling or mechanical force [6]. Hongqu and Xiaoqu are two kinds of commonly used Qu to brew rice wine. As the fermentation starter for Hongqu rice wine, Hongqu (also known as red yeast rice) is usually prepared from steamed rice by solid-state fermentation in an open environment, with the inoculation of this starter culture being abundant in microorganisms such as *Monascus*, *Rhizopus*, and *Aspergillus* [7,8]. Xiaoqu (also known as Yaoqu and Jiuyao) is a starter for saccharification and fermentation required for the brewing of special rice wine in Southern China [9]. It is made from rice flour (or rice bran) and Chinese herbal medicine as raw materials and fermented with the inoculation of mature koji produced in the previous year as the fermentation starter [10]. The dominant microorganisms in Xiaoqu are mainly *Streptococcus*, *Weissella*, *Pediococcus*, *Lactobacillus*, *Saccharomycopsis*, *Saccharomyces,* and *Rhizopus* [11]. The disparate microbial constitution between Hongqu and Xiaoqu elicits disparities in microbial composition and metabolomic attributes throughout rice wine’s brewing process, ultimately impacting the wine’s flavor nuances and comprehensive quality. However, there is no comparative study on the microbial and metabolomic profiles during the rice wine brewing process using Hongqu and Xiaoqu as fermentation starters.

Flavor is a pivotal aspect influencing the quality and consumer appeal of rice wine, and it is closely related to the metabolism of the microbial communities during fermentation. This intricate metabolic network involves dynamic interactions between various bacteria, molds, and yeasts [12,13]. It is necessary to conduct a more thorough investigation to clarify the ability of microbial communities to improve the flavor quality of rice wine. The protein in the raw material obtained from brewing rice wine is decomposed into amino acids through microbial metabolism and then transformed into alcohols and esters with special flavors under certain conditions. Higher alcohols are of great significance to the special flavor of rice wine [14]. A moderate amount of higher alcohols, especially less than 300 mg/L, is conducive to the smoothness of rice wine’s flavor and the balance of the taste [15]. However, it has been established that high levels of higher alcohols (>400 mg/L) adversely impact the taste profile and overall flavor quality of alcoholic beverages [16]. Drinking rice wine with a high concentration of higher alcohols may result in uncomfortable symptoms, such as headaches and hangovers [17,18]. Additionally, it may lead to potential health implications, including the risk of liver disease [19]. Throughout the fermentation of rice wine, microbial metabolism also produces a number of potentially harmful compounds, particularly biogenic amines (BAs), which are primarily synthesized by microorganisms through the decarboxylation of amino acids catalyzed by decarboxylase [20]. Different types of enterobacter and lactic acid bacteria contribute to the production of BAs [21,22]. It was previously documented that the concentration of biogenic amines in Chinese rice wine tends to be higher in comparison to other brewing wines, with an average content of up to 115 mg/L [23,24]. The consumption of rice wine with a high content of BAs increases the likelihood of adverse physiological effects, including headaches, an irregular heartbeat, and nausea [17,25,26]. In addition, alcohol has a notable suppressive effect on the activity of amine oxidase, so it is necessary to impose stricter limits on the content of BAs in alcoholic drinks [27]. Previously, we studied the relationship between BAs and microbiota during the production of Hongqu rice wine [11]. However, the microbiological mechanism of BA generation in the brewing process of Xiaoqu rice wine and its differences from Hongqu rice wine have not yet been determined.

Certain researchers employed a high-throughput sequencing (HTS) methodology to explore the disparities in microbial communities and volatile metabolites across various traditional starter cultures [9]. However, a microbiological analysis is often confined to the genus level, potentially rendering the outcomes of such studies incongruent with the actual conditions. Metagenomics offers a valuable perspective by elucidating the biological relevance of microbial species that harbor crucial genes in the brewing environment. In this study, we contrasted the temporal variations in the dynamics of BAs, higher alcohols, and volatile profiles in the brewing processes of rice wines fermented with Hongqu and Xiaoqu (HQW and XQW). Additionally, metagenomic sequencing technology was utilized for the analysis of the microbial community and functional genes in the fermentation process of HQW and XQW. The aim of this study was to clarify how microbial communities affect the flavor and safety quality of HQW and XQW, and the results may be helpful to improve the flavor quality and safety of rice wine.

## 2. Materials and Methods

### 2.1. Sample Collection

Hongqu and Xiaoqu were purchased from wine factories in Ningde city of Fujian province in China. HQW and XQW were fermented with glutinous rice as raw materials according to traditional production technology. Briefly, to initiate the brewing of HQW and XQW, glutinous rice was soaked in water (25 °C) overnight and then steamed at 100 °C. The steamed and cooled rice was placed into a wine jar and thoroughly mixed with 10% Hongqu or 1% Xiaoqu (wine starter) and 150% sterilized water. Subsequently, the wine jar was wrapped with eight layers of breathable gauze to ensure smooth fermentation, and it was then allowed to rest at a controlled temperature of 18 °C for ten days. It was then wrapped with a glass bottle gap for 30 days to facilitate anaerobic fermentation. The detailed method of sample collection was described in our prior research [15]. The supernatants were used for the determination of physicochemical parameters, biogenic amines, and volatile components, while the sediments were used for high-throughput sequencing of the microbial community. 

### 2.2. Physicochemical Parameter Determination

The alcohol concentrations in the HQW and XQW samples were measured using an Agilent 7890A gas chromatograph equipped with a hydrogen flame ionization detector and a J&W CP-Wax 57 CB capillary column (30 m × 0.25 mm × 0.25 μm) (Agilent Technologies, Little Falls, DE, USA). The operating parameters were determined by referring to the method described in our previous study [13]. Total acid content and amino nitrogen content were measured according to the Chinese National Standard method (GB/T 13662-2018 Rice Wine) [11]. The content of reducing sugars was determined by the dinitrosalicylic acid (DNS) method [28].

### 2.3. Higher Alcohol Determination

Agilent 7890A gas chromatograph (Agilent Technologies, Little Falls, DE, USA) was used to analyze and determine the contents of various higher alcohols in fermentation samples based on the standard curves of various higher alcohols. The detailed operational parameters refer to the method described in our prior research [29].

### 2.4. Biogenic Amine Determination

The content of biogenic amines (BAs) in the fermentation samples was determined using 1260 Infinity II HPLC instrumentation (Agilent Technologies, Little Falls, DE, USA) based on the Chinese standard GB 5009.208-2016 [30]. The detailed operational parameters refer to the method described in our prior study [11].

### 2.5. Volatile Flavor Compound Determination

Headspace solid-phase microextraction (HS-SPME) and gas chromatography–mass spectrometry (GC-MS) were used for qualitative and quantitative analyses of volatile components in the fermentation samples [31]. Detailed operating parameters of sample handling and GC-MS have been described in our previous studies and can be directly found in [15]. The internal standard method combined with the correction factor method was used for the quantitative analysis of each volatile substance, that is, the ratio of the peak area of each volatile substance to the peak area of the internal standard (2-octanol) of a known concentration combined with the correction factor was used to quantify the volatile component content, and then the odor activity value (OAV) of each volatile compound was calculated.

### 2.6. Microbial Metagenomic Sequencing

The CTAB-based method was used to extract the DNA of the microbiome from Hongqu, Xiaoqu, and the fermented sample (*Jiupei*) was collected at days 2, 5, 10, 20, and 30 [32]. NanoDrop2000 (Thermo Fisher Scientific, Wilmington, NC, USA) was used to detect the quality and purity of the extracted DNA, respectively. The checked genomic DNA was sent to metagenomic sequencing based on the Illumina NovaSeq Xten (Illumina Inc., San Diego, CA, USA) [33]. The metagenomic data were analyzed in depth according to the methods reported in the previous literature [34]. High-quality short reads for each DNA sample were assembled using MetaVelvet [35]. Sequenced raw data were deposited in the Sequence Read Archive of the National Center for Biological Information (NCBI) under the accession number PRJNA1102276. KEGG database was used to explore the functional genes and metabolic pathways closely related to BAs and flavor component metabolism.

### 2.7. Statistical Analysis and Visualization

Heat maps of volatile flavor compounds and functional gene bubble maps were drawn using R software (version 3.3.3). Principal component analysis (PCA) was performed using SIMCA-14.1 software (Umetrics AB, Umea, Vasterbotten, Sweden). STAMP software (version 2.1.3) was used to map the differences in taxonomic abundance of microorganisms. Subsequently, the correlations between the key microbial species and BAs as well as the characteristic volatile components were calculated based on Spearman correlation and visualized through R software (Ver. 3.3.3) with Psych, Reshape2, and heatmap packages.

## 3. Results and Discussion

### 3.1. Dynamics of Physicochemical Parameters during HQW and XQW Brewing Processes

As depicted in Figure 1, in the early stage of the HQW brewing process (day 1–day 7), the reducing sugar concentration showed a sharp increasing trend, reached a peak (155.17 mg/mL) on the seventh day, and a sharp downward trend followed. In contrast, the concentration of reducing sugar did not show a sharp increase during the XQW brewing process, although it increased slightly on the third day of the fermentation process and then displayed a slow decline. The high starch substrate in the initial stage of fermentation is conducive to the growth and reproduction of amylase-producing microorganisms such as *Aspergillus* and *Rhizopus*, and it promotes the hydrolysis of starch to reduce sugar. As the fermentation progresses, the reducing sugar is gradually converted by the yeast into alcohol, which, in turn, inhibits the activity of the amylase-producing microorganisms [36,37]. Consequently, the content of reducing sugar exhibited a decreasing trend. At the early stage of brewing, the reducing sugar content was observed to be higher in HQW compared to XQW, indicating that the microorganisms in the HQW brewing process had better liquefaction and saccharification capabilities, or that the microorganisms in the XQW brewing process had more powerful glycolysis capabilities. During the early and middle stages of fermentation (day 1–day 20), the alcohol content in both HQW and XQW showed a gradually increasing trend. In contrast, in the late stage of fermentation (after day 20), the microorganisms in *Jiupei* consumed a large amount of nutrients, the growth activity of the yeast was inhibited, and the efficiency of the conversion of reducing sugar to alcohol was greatly reduced, at which time the alcohol yield reached its maximum [38]. It is noteworthy that the alcohol concentration of HQW was lower than that of XQW when fermentation was completed. As a key index of the fermentation of Chinese rice wine, the acidity reflects and affects the growth and metabolism of microorganisms in the fermentation process. From the perspective of drinking comfort, the acidity of Chinese rice wine was best when it was no higher than 10 g/L. At present, the acidity control of the Chinese rice wine brewing process is a big problem in the industry, and some types of Qu easily lead to high acidity. It was observed that both HQW and XQW experienced a period of significant increase in total acidity during the initial stage of the rice wine brewing process, followed by a stabilization of the total acidity in the later stage of brewing. At the end of fermentation, the total acidity of XQW reached 10.54 g/L, which was significantly higher than that of HQW (5.59 g/L). Amino acids give rice wine taste characteristics, such as umami, sweet, bitter, and astringent. Additionally, amino acids are important nutrients and flavor precursors, providing nitrogen sources for microbial growth during fermentation [39]. As a key component, amino acids are mainly derived from the hydrolysis of proteins in raw materials. Taking rice wine brewing as an example, glutinous rice, as one of the commonly used raw materials, is rich in protein. During the brewing process of rice wine, protease and carboxypeptidase secreted by *Aspergillus* and *Monascus* act on the proteins in glutinous rice to produce amino acids or oligopeptides. This may be the reason for the increase in amino acid nitrogen during the rice wine brewing process. The abundance heat map of the amino acid contents in HQW and XQW is shown in Appendix A. In this study, the content of amino nitrogen during the HQW and XQW brewing processes increased with fermentation, and the content of amino nitrogen in HQW was higher than that in XQW at the end of brewing.

### 3.2. Dynamics of Higher Alcohols during HQW and XQW Brewing Processes

Higher alcohols commonly found in rice wine have either a positive or negative effect on the sensory flavor of rice wine depending on the concentration of higher alcohols in rice wine [40]. Therefore, the changes in higher alcohols during the brewing processes of HQW and XQW was investigated, and it was found that the trend in higher alcohol production was similar for HQW and XQW, with a significant increase in the alcohol content at the early stage of fermentation, which reached a plateau in HQW after the 10th day of fermentation and in XQW after the 20th day of fermentation (Figure 2). At the end of brewing, the content of higher alcohols in XQW was 685.30 mg/L, which was significantly higher than that in HQW (318.53 mg/L). However, a high level of higher alcohols (>400 mg/L) was proven to negatively affect the flavor quality of alcoholic beverages [16]. Therefore, the content of higher alcohols in the brewing process of XQW should be controlled. At the end of brewing, the contents of isobutanol, 3-methyl-1-butanol, and phenethyl alcohol in XQW were 2.8, 2.6, and 2.9 times higher than those in HQW, respectively. Unexpectedly, the content of n-propanol in HQW (78.60 mg/L) was significantly higher than that in XQW (32.98 mg/L). Meanwhile, we found that the content of higher alcohols in HQW showed a decreasing trend from the 30th to 45th day of brewing, which might be due to the generation of corresponding esters by higher alcohols and organic acids under the action of esterifying enzymes produced by yeasts [41].

### 3.3. Dynamics of Biogenic Amines during HQW and XQW Brewing Processes

Rice wine contains a large amount of BAs. An appropriate amount of BAs has a positive effect on promoting human growth, enhancing metabolic vitality, and clearing free radicals in the body. However, when the human body takes in too much BAs from foods, blood vessels and arteries will relax, resulting in migraines and other hazards [42,43]. As can be seen in Figure 3, the BA contents in the HQW and XQW brewing processes showed a gradual increasing trend during the whole fermentation process, although there were certain fluctuations. At the end of fermentation, the BA content in HQW was significantly higher than that in XQW (65.61 mg/L and 37.48 mg/L, respectively). Among them, the contents of BAs in HQW fluctuated from the 5th to the 10th day of fermentation, which might be due to the production of amine oxidase by some microorganisms to degrade BAs [27]. In addition, putrescine is the most abundant BA in HQW [44], which is mainly produced at the early stage of fermentation. The concentrations of putrescine in HQW and XQW at the end of fermentation were 46.10 mg/L and 8.83 mg/L, respectively. In addition, the content of spermidine in HQW was slightly higher than that in XQW. However, tyramine was only detected in XQW, and its content in XQW was 10.54 mg/L at the end of fermentation. There were no significant differences in the levels of cadaverine and spermine between HQW and XQW. At present, some countries only set a limit standard for the histamine content in wine, among which the highest limit standard is 10 mg/L in Australia and Switzerland and 8 mg/L in France, and the most stringent requirement is 2 mg/L in Germany [45]. Notably, histamine was not detected in HQW and XQW. Furthermore, according to the recommendations of the European Food Safety Authority (EFSA), the daily intake of tyramine should not exceed 800 mg/kg [46].

### 3.4. Dynamics of Volatile Organic Components during HQW and XQW Brewing

The composition and abundance of volatile organic components (VOCs) serve as crucial indicators for evaluating the flavor quality of rice wine. As shown in Figure 4A and Table 1, a total of 101 VOCs were detected in XQW and HQW. As fermentation proceeded, the content of volatile organic compounds in XQW and HQW increased obviously (Figure 4A). It is worth noting that the abundances of VOCs, such as benzyl alcohol [C65], ethyl stearate [C95], benzaldehyde [C26], decanal [C24], nonanal [C17], and hexadecane [C34], were enriched in the early stage of XQW brewing but decreased in the late stage of XQW brewing. 1-Pentanol [C7], 2-methylpyrazine [C9], 1-hexanol [C14], and (2E)-2-octen-1-ol [C36] were enriched in the early stage of HQW brewing. Ethyl laurate [C60], ethyl linoleate [C98], ethyl octanoate [C19], ethyl butyrate [C2], ethyl decanoate [C38], ethyl nonanoate [C27], 2-nonanol [C25], nonanoic acid [C86], acetic acid [C22], ethyl 9-decanoate [C44], ethyl caproate [C6], and ethyl heptanoate [C13] were enriched in the late stage of HQW brewing. Ethyl acetate [C1], dodecanoic acid [C97], 1-nonanol [C41], ethyl 2-phenylacetate [C55], ethyl myristate [C77], phenethyl acetate [C58], 1-decanol [C52], isoamyl acetate [C4], citronellol [C53], gamma-hydroxybutyric acid [C37], ethyl 9-hexadecenoate [C91], isobutyl octyl phthalate [C99], butyl ethyl succinate [C56], ethyl isopentyl succinate [C69], 2-ethylbenzaldehyde [C57], ethyl glutarate [C54], and geraniol [C61] in XQW were significantly higher than HQW.

A principal component analysis (PCA) can better reflect the difference in metabolic characteristics between the two groups of samples, so we used a PCA to analyze the change trend in VOCs in the brewing processes of HQW and XQW. With the progress of fermentation, the two kinds of rice wine brewing samples were gradually separated on the PCA score scatter plot, indicating that there was a significant difference in the VOC composition during the HQW and XQW brewing processes (Figure 4B). Specifically, fermentation samples from the early stage of the HQW and XQW brewing process were primarily concentrated in the third quadrant of the PCA score scatter plot. Starting on the third day of the fermentation process, XQW underwent a positional change, transitioning from the third quadrant to the fourth. In contrast, HQW displayed a different pattern, moving from the third quadrant to the first quadrant by the seventh day of fermentation. In contrast, the sample of the mash derived from the ultimate phase of the process demonstrates that XQW brewing (XQW_D45) was mainly characterized by 1-nonanol [C41], ethyl 2-phenylacetate [C55], isobutyl octyl phthalate [C99], geraniol [C61], 3-methyl-1-butanol [C5], phenylethyl alcohol [C70], and so on. Utilizing a hierarchical clustering analysis (HCA), as depicted in Figure 4D, the categorization of the XQW and HQW mash samples revealed three distinct clusters, group I (XQW_D3-D45, group II (HQW_D7-D45), and group III (XQW_D1-D2, HQW_D1-D5), suggesting that there was a significant difference in the composition of VOCs between the two rice wines. At the early stage of fermentation, rice wine was in the stage of saccharification, and the content of volatile flavor substances in rice wine was low [29], so the volatile substances of XQW_D1-D2 and HQW_D1-D5 were similar. At the later stage of brewing, with the growth and metabolism of different microorganisms in XQW and HQW, a large number of volatile flavor substances was produced. There was a significant difference in the composition of VOCs produced during the brewing processes of HQW and XQW.

The odor activity value (OAV) serves as a metric, indicating the proportion of a VOC’s concentration to its sensory detection threshold, which is widely employed to evaluate the significance of VOCs in contributing to a sample’s aroma profile [47]. VOCs with OAVs > 1 are typically recognized as pivotal factors contributing to the characteristic taste and comprehensive aroma signature of rice wine [48]. In this study, characteristic aroma-active compounds between HQW and XQW were identified by calculating the OAV of each volatile compound. As demonstrated in Table 1, 23 and 19 VOCs with OAVs > 1 were found in XQW and HQW, respectively. Amongst the VOCs present in rice wine fermented with Xiaoqu (XQW), 2-methoxy-4-vinylphenol [C87] displays the highest OAV (OAV of 517.07 with a clove and smoky aroma), followed by cis-6-dodecen-4-olide [C94] (OAV of 153.37 with a sweet, fruity aroma), 3-(methylsulfanyl)-1-propanol [C47] (OAV of 109.93), decanal [C24] (OAV of 34.95), benzeneacetaldehyde [C39] (OAV of 13.24), ethyl palmitate [C88] (OAV of 11.48), and so on. However, in rice wine fermented with Hongqu (HQW), ethyl octanoate [C19] displays the highest OAV (OAV of 611.27 with fruity smells), followed by decanal [C24] (OAV of 55.41 with citrus, orange peel, and green melon nuance smells), 3-(methylsulfanyl)-1-propanol [C47] (OAV of 32.03), nonanal [C17] (OAV of 23.91), ethyl caproate [C6] (OAV of 23.31), and 5-pentyldihydro-2(3H)-furanone [C75] (OAV of 20.19). In particular, ethyl caprylate [C19] also had the highest OAV in Fenjiu, Baofengjiu, and Qingkejiu, showing a strong fruit aroma [47].

**Table 1 foods-13-02452-t001:** Characteristic volatile flavor components in rice wine fermented with Xiaoqu (XQW) and rice wine fermented with Hongqu (HQW).

Compound	CAS	Concentration (ug/L)		*p*-Value	Threshold (mg/L) ^#^	OAV	
		XQW-D45	HQW-D45			XQW-D45	HQW-D45
Acids							
Acetic acid [C22]	64-19-7	0.00	59.35	0.00180	60	0.00	0.00
γ-Hydroxybutyric acid [C37]	591-81-1	11.66	6.78	0.00869	-	-	-
Hexanoic acid [C63]	142-62-1	41,157.38	59,967.30	0.00104	5	8.23	11.99
Heptanoic acid [C72]	111-14-8	0.00	2415.11	0.00002	2.3	0.00	1.05
Octanoic acid [C78]	124-07-2	138.67	349.39	0.00267	0.91	0.15	0.38
Nonanoic acid [C86]	112-05-0	0.00	1865.22	0.00261	3	0.00	0.62
n-Decanoic acid [C89]	334-48-5	6481.45	11,408.41	0.10549	10	0.65	1.14
Myristic acid [C100]	544-63-8	4.19	8.71	0.01127	10	0.00	0.00
n-Hexadecanoic acid [C101]	57-10-3	17.51	27.97	0.03168	-	-	-
Alcohols							
Isobutanol [C3]	78-83-1	56,372.81	21,780.91	0.02243	100	0.56	0.22
3-Methyl-1-butanol [C5]	123-51-3	166,518.03	96,710.48	0.00073	50	3.33	1.93
1-Pentanol [C7]	71-41-0	0.00	258.04	0.00095	50	0.00	0.01
2-Heptanol [C12]	543-49-7	0.00	54.04	0.00012	0.25	0.00	0.22
1-Hexanol [C14]	111-27-3	770.90	881.43	0.06357	4	0.19	0.22
3-Ethoxy-1-propanol [C15]	111-35-3	2.39	6.21	0.00019	10	0.00	0.00
1-Octen-3-ol [C20]	3391-86-4	19.35	36.38	0.00125	0.2	0.10	0.18
1-Heptanol [C21]	111-70-6	66.05	94.95	0.00797	1	0.07	0.09
2-Ethylhexanol [C23]	104-76-7	68.55	60.72	0.06074	0.83	0.08	0.07
2-Nonanol [C25]	628-99-9	3.73	49.45	0.00001	0.075	0.05	0.66
(2R,3R)-(−)-2,3-Butanediol [C28]	24347-58-8	0.00	34.40	0.02356	400	0.00	0.00
Linalool [C30]	78-70-6	5.94	0.00	0.00000	0.08	0.07	0.00
1-Octanol [C31]	111-87-5	115.13	61.49	0.00004	0.9	0.13	0.07
Isopropanol [C32]	67-63-0	0.00	17.77	0.01262	100	0.00	0.00
(2E)-2-Octen-1-ol [C36]	18409-17-1	6.94	0.00	0.00198	0.04	0.17	0.00
1-Nonanol [C41]	143-08-8	85.12	55.00	0.00048	0.08	1.06	0.69
(3Z)-3-Nonen-1-ol [C43]	10340-23-5	15.84	3.08	0.00000	-	-	-
(+)-Borneol [C46]	507-70-0	6.65	0.00	0.00000	-	-	-
3-(Methylsulfanyl)-1-propanol [C47]	505-10-2	3957.65	1152.95	0.00001	0.036	109.93	32.03
1-Decanol [C52]	112-30-1	17.37	8.67	0.00041	0.18	0.10	0.05
Geraniol [C61]	106-24-1	13.28	0.00	0.01464	0.036	0.37	0.00
Phenylethyl alcohol [C70]	60-12-8	45,312.47	27,918.40	0.00003	5	9.06	5.58
2,4,7,9-Tetramethyl-5-decyn-4,7-diol [C79]	126-86-3	60.09	3.56	0.00035	-	-	-
3,4-Dimethylbenzyl alcohol [C82]	6966-10-5	6.73	0.00	0.00005	-	-	-
(+)-Cedrol [C83]	77-53-2	15.91	13.02	0.40790	-	-	-
Aldehydes							
Octanal [C11]	124-13-0	0.00	28.02	0.09063	0.04	0.00	0.70
Nonanal [C17]	124-19-6	72.02	358.61	0.17653	0.015	4.80	23.91
Decanal [C24]	112-31-2	174.76	277.04	0.16765	0.005	34.95	55.41
Benzaldehyde [C26]	100-52-7	15.40	23.56	0.00021	1.5	0.01	0.02
Myristaldehyde [C35]	124-25-4	0.00	8.92	0.00599	100	0.00	0.00
Benzeneacetaldehyde [C39]	122-78-1	1324.39	1054.84	0.56011	0.1	13.24	10.55
3,4-Dimethylbenzaldehyde5973-71-7 [C50]	5973-71-7	1.96	0.00	0.00007	-	-	-
2-Ethylbenzaldehyde [C57]	22927-13-5	517.06	421.59	0.21568	-	-	-
Esters							
Ethyl acetate [C1]	141-78-6	115,127.14	88,800.06	0.05713	30	3.84	2.96
Ethyl butyrate [C2]	105-54-4	142.71	237.50	0.40513	0.4	0.36	0.59
Isoamyl acetate [C4]	123-92-2	781.52	948.14	0.04757	0.68	1.15	1.39
Ethyl caproate [C6]	123-66-0	1322.60	4895.86	0.00000	0.21	6.30	23.31
Ethyl N-benzoylglycinate [C10]	1499-53-2	10.96	20.81	0.00038	-	-	-
Ethyl heptanoate [C13]	106-30-9	0.00	434.26	0.00014	0.4	0.00	1.09
Ethyl octanoate [C19]	106-32-1	1478.22	3056.34	0.00035	0.005	295.64	611.27
Ethyl nonanoate [C27]	123-29-5	0.00	99.90	0.00002	1.2	0.00	0.08
Ethyl 2-hydroxyhexanoate [C29]	6946-90-3	33.83	53.59	0.00045	-	-	-
Ethyl decanoate [C38]	110-38-3	3315.39	5265.61	0.04238	1.5	2.21	3.51
Diethyl succinate [C42]	123-25-1	982.12	1525.91	0.00011	75	0.01	0.02
Ethyl 9-decenoate [C44]	67233-91-4	0.00	9.46	0.00031	-	-	-
Ethyl glutarate [C54]	818-38-2	16.15	4.75	0.00044	-	-	-
Ethyl 2-phenylacetate [C55]	101-97-3	230.17	139.08	0.00011	0.15555	1.48	0.89
Butyl ethyl succinate [C56]	1000324-85-1	28.45	16.08	0.00017	-	-	-
Phenethyl acetate [C58]	103-45-7	377.34	560.37	0.00030	2	0.19	0.28
Ethyl laurate [C60]	106-33-2	15.06	13.21	0.40599	3.5	0.00	0.00
n-Butyl butanoate [C64]	109-21-7	133.20	33.95	0.00001	0.087	1.53	0.39
Cyclopentyl butyrate [C66]	6290-13-7	5.78	1.99	0.00030	-	-	-
Ethyl 3-phenylpropanoate [C67]	2021-28-5	6.07	0.00	0.00000	0.0016	3.79	0.00
Ethyl 3-hydroxydodecanoate [C68]	183613-15-2	31.76	66.73	0.00070	-	-	-
Ethyl isopentyl succinate [C69]	28024-16-0	47.86	29.84	0.00617	-	-	-
Ethyl myristate [C77]	124-06-1	23.57	3.55	0.00098	0.18	0.13	0.02
Ethyl 3-hydroxytridecanoate [C80]	107141-15-1	23.44	34.10	0.14882	-	-	-
Diethyl suberate [C81]	2050-23-9	9.55	11.91	0.24807	-	-	-
Ethyl palmitate [C88]	628-97-7	17,223.92	5502.63	0.00292	1.5	11.48	3.67
Ethyl 9-hexadecenoate [C91]	54546-22-4	16.09	0.00	0.00000	-	-	-
Cis-6-Dodecen-4-olide [C94]	18679-18-0	15.34	0.00	0.00000	0.0001	153.37	0.00
Ethyl stearate [C95]	111-61-5	142.08	29.85	0.00027	15000	0.00	0.00
Ethyl oleate [C96]	111-62-6	1346.04	465.10	0.00970	3.5	0.38	0.13
Dodecanoic acid [C97]	143-07-7	12.19	13.67	0.69195	0.5	0.02	0.03
Isobutyl octyl phthalate [C99]	1000309-04-5	43.27	24.98	0.02660	-	-	-
Ketones							
2-Nonanone [C16]	821-55-6	5.70	5.96	0.13240	0.2	0.03	0.03
Isophorone [C33]	78-59-1	2.55	7.08	0.00075	0.1	0.03	0.07
Acetophenone [C40]	98-86-2	15.81	8.81	0.00122	3	0.01	0.00
Geranylacetone [C62]	3796-70-1	90.00	139.19	0.01801	0.06	1.50	2.32
5-Pentyldihydro-2(3H)-furanone [C75]	104-61-0	0.00	195.85	0.00001	0.0097	0.00	20.19
Others							
Hexadecane [C34]	544-76-3	9.05	6.92	0.28655	300	0.00	0.00
Heptadecane [C45]	629-78-7	10.66	4.28	0.00596	-	-	-
Naphthalene [C49]	91-20-3	5.08	2.11	0.00385	0.36	0.01	0.01
Trans-Anethole [C59]	4180-23-8	21.55	0.00	0.00001	0.015	1.44	0.00
Benzothiazole [C71]	95-16-9	80.62	41.48	0.00164	0.08	1.01	0.52
2,3-Dihydro-1-benzofuran [C93]	496-16-2	19.32	7.69	0.00344	-	-	-
Phenols							
4-(Methylsulfanyl)phenol [C51]	1073-72-9	11.46	3.98	0.00441	0.8	0.01	0.00
Methyleugenol [C74]	93-15-2	23.44	0.00	0.00001	1.25	0.02	0.00
4-Ethylguaiacol [C76]	2785-89-9	227.57	0.00	0.00003	0.07	3.25	0.00
Chavibetol [C84]	501-19-9	231.37	0.00	0.00006	-	-	-
4-Ethylphenol [C85]	123-07-9	186.66	0.00	0.00008	0.3	0.62	0.00
2-Methoxy-4-vinylphenol [C87]	7786-61-0	1551.22	12.67	0.00003	0.003	517.07	4.22
2,4-Bis(2-methyl-2-propanyl)phenol [C90]	96-76-4	452.19	362.51	0.44719	0.5	0.90	0.73
Pyrazines							
2-Methylpyrazine [C9]	109-08-0	0.00	191.91	0.00596	100	0.00	0.00

^#^ Thresholds were obtained from references [49,50,51,52] or *Compilations of Odour Threshold Values in Air, Water and Other media* [53]. The concentration of volatile flavor substances in the table is the average value of three parallel experiments.

### 3.5. Dynamics of Microbial Community during HQW and XQW Brewing Processes

Differences in the microbial community composition may account for the significant differences in the flavor characteristics between HQW and XQW. Therefore, the dynamic changes in the microbial community during the HQW and XQW brewing processes were analyzed through a microbiome analysis based on metagenomic sequencing. As depicted in Figure 5A, the stacked histogram of the major bacterial genera revealed that *Weissella*, *Pantoea*, *Pediococcus*, *Enterobacter*, *Cronobacter*, *Lactobacillus,* and *Kosakonia* were predominant in XQW brewing. HQW’s major bacterial genera included *Weissella*, *Burkholderia*, *Pantoea*, *Enterobacter*, *Klebsiella,* and *Leuconostoc*. Among them, *Weissella*, *Leuconostoc*, *Lactobacillus,* and *Pediococcus* are categorized as lactic acid bacteria (LAB), suggesting that LAB might play a role in the development of aroma constituents during rice wine fermentation, which aligned with the findings in our previous study [11]. Notably, *Bacillus*, *Lactobacillus*, and *Weissella* were also identified as the primary bacterial genera playing crucial roles in the fermentation of Daqu light-aroma Baijiu [54]. As the brewing processes progressed, the relative abundances of *Enterobacter*, *Klebsiella*, *Leuconostoc,* and *Kosakonia* decreased in the HQW and XQW brewing processes. On the contrary, the relative abundance of *Weissella* rose to 45.19% in the late stage of the HQW brewing process but decreased to 13.77% at the end of the XQW brewing process. Although the relative abundance of *Pediococcus* was extremely low in Xiaoqu, its abundance gradually increased during the brewing process of XQW, especially in the late stage of brewing when its relative abundance increased to 62.25%. Figure 5B illustrates the dynamic evolution of the major fungal communities at the genus level during the XQW and HQW brewing processes. The most prominent members of a fungal community in the HQW brewing process were *Saccharomyces* and *Monascus*, which dominated the whole brewing process. The relative abundance of *Saccharomyces* in the HQW brewing process accounted for more than 75% of the entire fungal community, whereas its relative abundance was only about 20% in the XQW brewing process. The relative abundance of *Monascus* decreased from 87.54% to 16.56% during the HQW brewing process, but its abundance in the XQW brewing process was so low that it was undetectable. It has been shown that *Saccharomyces* and *Monascus* competed with each other during fermentation and that the growth metabolism of *Monascus* was inhibited with ethanol accumulation [39]. In the early stage of aerobic fermentation, the brewing system contained a large amount of glutinous rice starch, which was favorable for the growth of *Monascus*, while the increase in the alcohol content on the fifth day of fermentation was unfavorable for the growth of *Monascus*, so its relative abundance decreased substantially. *Saccharomyces*, on the other hand, has been shown to be a major participant in alcoholic fermentation, as it is capable of converting sugars into ethanol and carbon monoxide [55]. In addition to *Saccharomyces*, *Rhizopus*, *Ascoidea*, *Wickerhamomyces*, *Pachysolen,* and *Cyberlindnera* were identified as the predominant fungal genera in the XQW brewing process, but they were hardly found in the HQW brewing process. Metagenomic sequencing showed that the relative abundance of *Rhizopus* in Xiaoqu was as high as 32.14%, and its abundance rose to 80.69% on the second day of the XQW brewing process and remained around 50% thereafter. *Rhizopus* and *Saccharomyces* play diverse roles in shaping the aroma and taste profile of rice wine at different stages of brewing. It is well known that *Rhizopus* and *Monascus* play a key role in the rice wine brewing process because they can secrete higher amounts of α-amylase and glucamylase [56], while *Saccharomyces*, *Wickerhamomyces,* and *Cyberlindnera* play an essential role in the production of alcohols and esters [57].

The dynamic changes in the dominant bacterial community at the species level during the XQW and HQW brewing processes are shown in Figure 6A. *Pediococcus pentosaceus*, *Weissella confusa*, unclassified_g_*Pantoea*, *Pantoea dispersa,* and unclassified_g_*Pediococcus* were the predominant bacterial species in the XQW brewing process, while *Burkholderia gladioli*, *Weissella cibaria*, unclassified_g_*Pantoea*, unclassified_g_*Burkholderia*, *Pantoea dispersa,* and unclassified_g_*Weissella* were the predominant bacterial species in the HQW brewing process. The composition of the dominant fungal community at the species level during the XQW and HQW brewing processes is shown in Figure 6B. The predominant fungal species in the XQW brewing process included *Saccharomyces cerevisiae*, *Rhizopus delemar*, *Pachysolen tannophilus*, *Rhizopus microsporus*, *Ascoidea rubescens*, *Wickerhamomyces anomalus*, *Wickerhamomyces ciferri*, *Rhizopus stolonifera,* and *Rhizopus oryzae*. As for the HQW brewing process, the only two fungal species that are dominant in abundance were *Monascus purpureus* and *Saccharomyces cerevisiae*. As evident from Figure 6B, the relative abundance of *Monascus purpureus* in Hongqu was as high as 90.94%, and its relative abundance in the HQW brewing process decreased significantly as fermentation progressed, reaching 16.42% on the 30th day of fermentation. In contrast, the relative abundance of *Saccharomyces cerevisiae* in Hongqu was only 0.12%, but its relative abundance was as high as 79.65% in the late stage of the HQW brewing process, while its relative abundance in the XQW brewing process was only 27% at the highest point. *Saccharomyces cerevisiae* was reported to be important for alcohols and other flavor compounds in fermentation [58]. In addition, *Saccharomyces cerevisiae* promoted the conversion of aroma components from alcohols and acids to esters during the solid-state analog fermentation of white wine [59].

To gain a deeper insight into the differences in the microbial composition between HQW and XQW, we employed the STAMP software (Ver. 2.1.3) to analyze and visualize the differential microbial taxa between HQW and XQW. The microorganisms with significant differences in relative abundance between HQW and XQW included 56 bacterial and 53 fungal species (Figure 7). *Weissella cibaria*, *Burkholderia gladioli*, unclassified_g_*Burkholderia*, unclassified_g_*Weissella*, *Klebsiella pneumoniae*, *Ralstonia solanacearum*, *Leuconostoc kimchii*, *Pseudomonas aeruginosa,* and *Pantoea ananatis* were more abundant in the HQW brewing process (Figure 7A), while *Pediococcus pentosaceus*, *Weissella confusa*, unclassified_g_*Pediococcus*, *Cronobacter dublinensis*, *Cronobacter sakazakii*, *Enterobacter roggenkampii*, *Lactobacillus plantarum*, *Pediococcus acidilactici*, *Cronobacter malonaticus,* and *Lactococcus lactis* were more abundant in the *XQW* brewing process (Figure 7A). Figure 7B shows the difference in the abundance of fungal communities at the species level between the XQW and HQW brewing processes. It can be seen that the abundance of most of the fungal species in the HQW brewing process was significantly lower than that in the XQW brewing process. Microbial species such as *Rhizopus delemar*, *Ascoidea rubescens*, *Rhizopus microsporus*, *Rhizopus stolonifer*, *Pachysolen tannophilus*, *Wickerhamomyces anomalus*, *Wickerhamomyces ciferrii*, *Rhizopus oryzae*, *Rhizopus azygosporus,* and *Cyberlindnera fabianii* were prominently enriched in the XQW brewing process, while *Saccharomyces cerevisiae*, *Saccharomyces pastorianus*, and *Saccharomyces boulardii* were significantly enriched in the HQW brewing process. Furthermore, *Rhizopus* exhibited a remarkable ability to synthesize a diverse range of flavor esters, such as butyl acetate, hexyl acetate, and geranyl acetate, demonstrating its efficiency in flavor production [60]. Additionally, *Wickerhamomyces* exhibited significant potential for ester production during simulated Baijiu fermentation [61].

### 3.6. The Correlations between Microbial Communities and Metabolites

The development of the flavor quality in rice wine is intimately linked to the composition and metabolic activities of the microbial community throughout the brewing process. Therefore, this study analyzed the correlations between the microbial phylotypes and metabolites in the HQW and XQW brewing processes through Spearman’s correlation analysis and identified the characteristic microbial species highly correlated with volatile flavor components, biogenic amines, and higher alcohols (Figure 8). As shown in Figure 8A, 3-methyl-1-butanol [C5], phenylethyl alcohol [C70], isobutanol [C3], ethyl glutarate [C54], and ethyl 3-phenylpropanoate [C67] were positively correlated with bacterial species such as *Leuconostoc mesenteroides*, *Lactobacillus sakei*, *Lactobacillus brevis*, *Lactobacillus plantarum,* and *Pediococcus pentosaceus* but negatively correlated with bacterial species such as *Salmonella enterica*, *Pantoea ananatis*, *Klebsiella pneumoniae,* and *Escherichia coli* (Figure 8A). It has been reported that *Leuconostoc* existed in various fermented foods, such as pickles, pickled peppers, and Sichuan dishes, and had a significant positive correlation with volatile compounds (VOCs) [62], which was consistent with the results of this study. According to a previous study, a variety of lactic acid bacteria (LAB), including *Streptococcus* and *Lactococcus*, may be the main contributors to biogenic amines during rice wine fermentation [11]. Spearman’s correlation analysis of the bacterial community and biogenic amines revealed that tyramine showed a positive correlation with *Pediococcus acidilactici*, *Pediococcus pentosaceus*, *Lactobacillus plantarum*, *Lactobacillus brevis*, *Lactobacillus sakei*, *Leuconostoc mesenteroides*, and *Enterococcus faecium* but showed a negative correlation with bacterial species such as *Lactococcus garvieae*, *Weissella cibaria*, etc. In particular, *Pediococcus acidilactici* M28, with a strong biogenic amine-degrading ability, was isolated from Chinese soybean paste and was able to degrade all eight kinds of biogenic amines [63]. Cadaverine was positively correlated with *Streptococcus pneumoniae*, *Weissella paramesenteroides*, *Weissella confuse,* and *Enterococcus faecium* but negatively correlated with bacterial species such as *Burkholderia vietnamiensis*, and *Burkholderia gladioli*.

The correlations between the characteristic fungal species and the metabolites in the HQW and XQW brewing processes are shown in Figure 8B. It was found that isobutanol [C3], 3-methyl-1-butanol [C5], and phenylethyl alcohol [C70] were positively associated with *Komagataella phaffi*, *Candida viswanathii*, *Pachysolen tannophilus*, *Kuraishia capsulata*, *Ogataea parapolymorpha,* and *Candida oxycetoniae*. In addition, putrescine was negatively associated with fungal species such as *Rhizopus oryzae*, *Mucor lusitanicus*, *Rhizopus microsporus*, *Rhizopus azygosporus*, *Mucor circinelloides*, *Rhizopus delemar*, and *Rhizopus stolonifer*. A reduction in bioamines by *Mucor racemosus* has been demonstrated [64]. However, tyramine was positively correlated with most of the fungal species such as *Rhizopus oryzae*, *Mucor lusitanicus*, *Rhizopus azygosporus*, *Mucor circinelloides*, *Komagataella phaffi*, *Candida viswanathii*, *Pachysolen tannophilus*, and *Candida glabrata*. Of course, the results of the statistical correlation analysis need to be verified by a fermentation experiment and metabolomics study. As the main characteristic volatile flavor component in XQW, 2-methoxy-4-vinylphenol [C87] showed positive correlations with bacterial species including *Pediococcus acidilactici*, *Pediococcus pentosaceus*, *Lactobacillus plantarum*, *Lactobacillus brevis*, *Lactobacillus sakei*, *Leuconostoc mesenteroides*, and *Enterococcus faecium* and most fungal species, such as *Komagataella phaffi*, *Candida viswanathii*, *Candida glabrata*, *Candida tropicalis*, *Debaryomyces fabryi*, *Pichia membranifaciens*, *Candida maltosa*, etc. Lactic acid bacteria, including *Lactobacillus*, *Pediococcus*, *Leuconostoc*, etc., being an essential component in fermentation processes, were categorized as a type of probiotic. They possess the ability to generate antibacterial and antioxidant effects [65].

### 3.7. Functional Genes and Microorganisms for Flavor and Biogenic Amine Metabolism

Based on the genes obtained from metagenomic sequencing, bubble plots of the abundance of functional genes corresponding to enzymes associated with flavor and BA metabolism were drawn (Figure 9). The classification of functional genes and the annotation results of microbial species are shown in Appendix A. Compared with HQW, the abundance of functional genes for glycogen phosphorylase [EC 2.4.1.1], glucan 1,4-alpha-glucosidase [EC 3.2.1.3], maltose phosphorylase [EC 2.4.1.8], oligo-1,6-glucosidase [EC 3.2.1.10], and maltogenic alpha-amylase [EC 3.2.1.133] were significantly higher in the XQW brewing process. Hexokinase [EC 2.7.1.1], glucokinase [EC 2.7.1.2], and polyphosphate glucokinase [EC 2.7.1.63] possess the enzymatic capacity to catalyze the conversion of α-D-glucose into α-D-glucose 6-phosphate, which, in turn, provides a rich substrate for subsequent glycolysis pathways. In addition, hexokinase [EC 2.7.1.1] exhibited a significantly higher abundance in XQW, primarily attributed to its derivation from *Rhizopus delemar*, *Ascoidea rubescens,* and *Hyphopichia burtonii*. Glucose-6-phosphate isomerase [EC 5.3.1.9] and 6-phosphofructokinase [EC 2.7.1.11] play a pivotal role in catalyzing the conversion of α-D-glucose-6-phosphate into β-D-fructose 1,6-bisphosphate, and they were primarily derived from *Pediococcus* and *Weissella cibaria*. Glycerate phosphomutase [EC 5.4.2.11] facilitates the conversion between glycerate-3P and glycerate-2P, and it was primarily derived from *Pediococcus*, *Weissella cibaria,* and *Pantoea* in the XQW brewing process.

Alcohol, ester, and phenolic flavor compounds are produced in rice wine fermentation under the action of the microbial community [66]. Higher alcohols are primarily generated through two distinct pathways: the degradation metabolic pathway (Ehrlich pathway) and the anabolic pathway (Harris pathway) [67]. The synthesis of n-propanol is mainly derived from glycerol phosphate and threonine. In the formation of n-propanol, propanediol dehydratase [EC 4.2.1.28] converts 1,2-propanediol to propanal, and propanediol dehydrogenase [EC 1.1.1.202] converts propanal to n-propanol. Through the microbial annotation of functional genes, we found that propanediol dehydratase [EC 4.2.1.28] and propanediol dehydrogenase [EC 1.1.1.202] were mainly derived from *Pediococcus pentosaceus*. Glycerol dehydrogenase [EC 1.1.1.6] converts hydroxyacetone to 1,2-propanediol, and it was mainly derived from *Weissella cibaria* in HQW and *Pantoea dispersa* in XQW. The abundance of gene-encoding aryl-alcohol dehydrogenase [EC 1.1.1.90, catalyzing phenylacetaldehyde to 2-phenylethanol] was significantly enriched in the XQW brewing process, and it was mainly derived from *Pediococcus pentosaceus* and *Weissella confusa*. Alcohol dehydrogenase [EC 1.1.1.1] and alcohol dehydrogenase (NADP+) [EC 1.1.1.2] are mainly responsible for participating in the conversion of aldehydes to alcohols [68]. In this study, we found that *Weissella cibaria*, *Pediococcus pentosaceus*, unclassified_g_*Burkholderia*, *Mixta calida,* and *Pantoea dispersa* carry the functional genes for these two enzymes. Pyruvate decarboxylase [EC 4.1.1.1] was an important carbon–carbon lyase in the anabolism of higher alcohols, and it can catalyze the formation of important precursors of higher alcohols, namely phenylacetaldehyde, isobutyraldehyde, and isovaleraldehyde. Pyruvate decarboxylase was more abundant in XQW, and it was mainly derived from *Saccharomyces cerevisiae* and *Rhizopus delemar*. These may be the reasons for the higher content of higher alcohols in XQW. Pyruvate decarboxylase is an important carbon–carbon lyase in the anabolism of higher alcohols, and it can catalyze the formation of important precursors of higher alcohols, namely phenylacetaldehyde, isobutyraldehyde, and isovaleraldehyde. The gene-encoding pyruvate decarboxylase was more abundant in the XQW brewing process, and it was mainly derived from *Saccharomyces cerevisiae* and *Rhizopus delemar*. In addition, functional genes corresponding to enzymes involved in valine biosynthesis, leucine biosynthesis, and phenylalanine biosynthesis (transaminase [EC 2.6.1.57], dihydroxy-acid dehydratase [EC 4.2.1.9], 3-isopropylmalate dehydratase [EC 4.2.1.33], 2-isopropylmalate synthase [EC 2.3.3.13], and chorismate mutase [EC 5.4.99.5]) were more abundant in the HQW brewing process. The microbial annotation of functional genes showed that *Burkholderia gladioli* and unclassified_g_*Pantoea* were the main contributors of the genes related to amino acid metabolism in the HQW and XQW brewing processes, respectively. Fatty acids are precursors of many characteristic flavor compounds (e.g., esters) in fermented foods, and their contents greatly affect the flavor quality of fermented foods [69]. The results of this study show that the abundance of gene-encoding enzymes related to fatty acid metabolism was higher during the XQW brewing process, such as enoyl-[acyl-carrier protein] reductase I [EC 1.3.1.9, oxidoreductases], fatty acid synthase [EC 2.3.1.86, acyltransferases], medium-chain acyl-[acyl-carrier-protein] hydrolase [EC 3.1.2.21, which produces long chain fatty acids], etc. The encoding genes of these enzymes were mainly derived from *Pediococcus pentosaceus*, *Rhizopus delemar*, unclassified_g_*Pediococcus*, etc. Notably, there were significant differences in the abundance of esters biosynthesis-related genes during the brewing of HQW and XQW. For example, alcohol dehydrogenase [EC 1.1.1.1, acting on the CH-OH group of donors], triacylglycerol lipase [EC 3.1.1.3, carboxylic-ester hydrolases], and carboxylesterase [EC 3.1.1.1, acting on ester bonds] are closely related to the biosynthesis of esters, and they were more abundant in the HQW brewing process. The microbial annotation of functional genes revealed that the gene-encoding alcohol dehydrogenase [EC 1.1.1.1] was mainly derived from *Weissella cibaria*, *Pediococcus pentosaceus,* and *Burkholderia gladioli* (Appendix A). In addition, *Burkholderia gladioli* and unclassified_g_*Burkholderia* were the main microbial species for carboxylesterase [EC 3.1.1.1]. Triacylglycerol lipase [EC 3.1.1.3] was primarily derived from *Hyphopichia burtonii* and *Enterobacter cloacae* in the XQW brewing process, while *Burkholderia gladioli* was derived from HQW. Alcohol O-acetyltransferase [EC 2.3.1.84] was primarily derived from *Saccharomyces cerevisiae*, which was consistent with the result of a previous study [56]. 2-Methoxy-4-vinylphenol [C87] is the characteristic volatile flavor component in XQW, which is produced mainly by the conversion of p-coumaric acid catalyzed by ferulic acid decarboxylase [EC 4.1.1.102] [70]. Based on the microbial annotation of functional genes (Appendix A), ferulic acid decarboxylase [EC 4.1.1.102] was primarily derived from *Saccharomyces cerevisiae* and *Monascus purpureus.* According to the function annotation, p-coumaric acid is derived from phenylalanine through the phenylalanine metabolic pathway [71]. However, it was suggested that phenylalanine and p-coumaric acid was probably mainly originated from raw materials rather than being produced through microbial metabolism during the rice wine brewing process [72].

Gene function annotation provides valuable insights into the regulatory mechanisms of BA metabolism during the HQW and XQW brewing processes. With agmatine as the starting substrate, putrescine synthesis can be further divided into two pathways, one of which is the formation of agmatine directly catalyzed by agmatinase [EC 3.5.3.11]. Secondly, agmatine is catalyzed by agmatine deiminase [EC 3.5.3.12] to form n-carbamoylputrescine, which then undergoes the action of n-carbamoylputrescine amidase [EC 3.5.1.53] to form putrescine [73]. The genes encoding the above three enzymes were mainly derived from *Burkholderia gladioli* and *Enterobacter* (Appendix A). Interconversion between putrescine, spermidine, and spermine is possible under suitable conditions. Putrescine can be degraded to spermidine by spermidine synthase [EC 2.5.1.16], and spermidine and spermine can be interconverted under the action of different enzymes [42]. Compared with XQW, the gene abundance of spermidine synthase was significantly higher in HQW, which may be the main reason for the relatively higher content of spermidine in HQW. The microbial annotation of functional genes indicated that spermidine synthase [EC 2.5.1.16] was mainly derived from unclassified_g_*Burkholderia* and *Burkholderia gladioli* in the HQW brewing process. Tyramine in rice wine was mainly produced by tyrosine under the catalytic action of tyrosine decarboxylase [EC 4.1.28], mainly derived from *Monascus* and *Ascoidea*. When the protein structure is destroyed, lysine decarboxylase [EC 4.1.1.18] reacts with L-lysine to produce cadaverine. As can be seen in Appendix A, the gene-encoding lysine decarboxylase [EC 4.1.1.18] was mainly derived from *Pantoea dispersa*, *Enterobacter cloacae*, and *Klebsiella pneumoniae* [74]. The precursor amino acids for the synthesis of BAs can be produced through the hydrolysis of proteins present in the winemaking raw material or synthesized by microbial metabolism during the winemaking process [75]. The results of the gene abundance analysis show that the abundances of genes related to amino acid synthesis in the brewing process of HQW were significantly higher than those of XQW, which could explain the higher contents of cademine, spermidine, and putrescine in the brewing process of HQW.

## 4. Conclusions

In this research, we compared the microbial ecology, BA contents, and volatile compounds during the HQW and XQW brewing processes. Our findings indicate that XQW contained more alcohols compared to HQW. Conversely, HQW exhibited a significantly higher content of BAs. The volatile flavor components of both HQW and XQW exhibited distinct profiles. Metagenomic sequencing further revealed that the brewing processes of HQW and XQW were governed by distinct microbial populations. Moreover, we identified the microbial genes responsible for the metabolism of BAs and higher alcohols in both rice wines. Notably, HQW possessed a higher abundance of genes related to BA synthesis and amino acid metabolism compared to XQW. Conversely, XQW exhibited a significantly higher abundance of genes linked to fatty acid metabolism. This comprehensive analysis provides crucial scientific insights into how microbial communities influence the flavor quality of HQW and XQW, paving the way for the selection of beneficial microorganisms to enhance the flavor profiles of rice wine. Therefore, a subsequent study can employ multi-omics integration technology for a more comprehensive and systematic analysis of the metabolic mechanism of BAs and higher alcohols.

## Figures and Tables

**Figure 1 foods-13-02452-f001:**
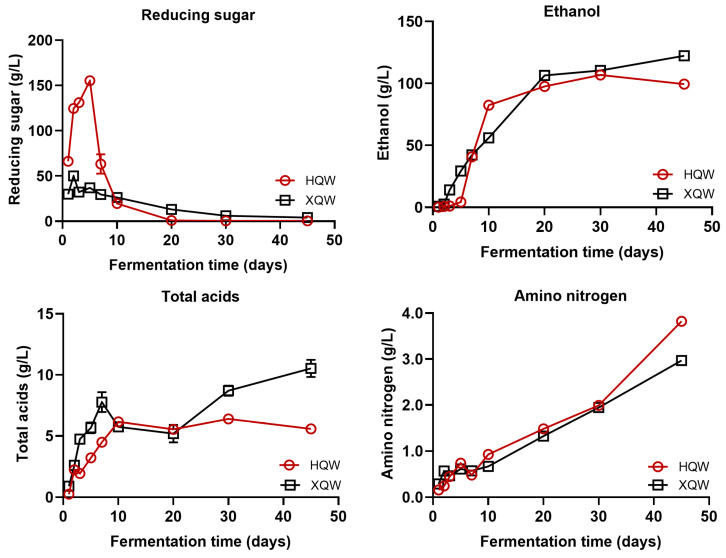
The dynamics of physicochemical parameters (including reducing sugar, ethanol, total acid, and amino nitrogen) during the HQW and XQW brewing processes.

**Figure 2 foods-13-02452-f002:**
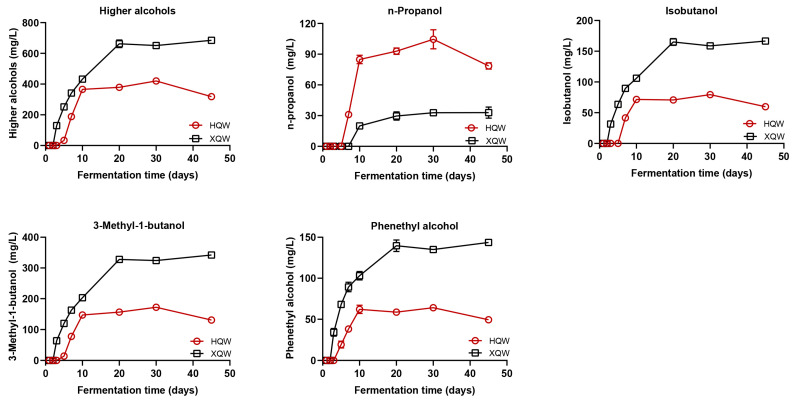
The dynamics of higher alcohols (including n-propanol, isobutanol, 3-methyl-1-butanol, and phenethyl alcohol) during the HQW and XQW brewing processes.

**Figure 3 foods-13-02452-f003:**
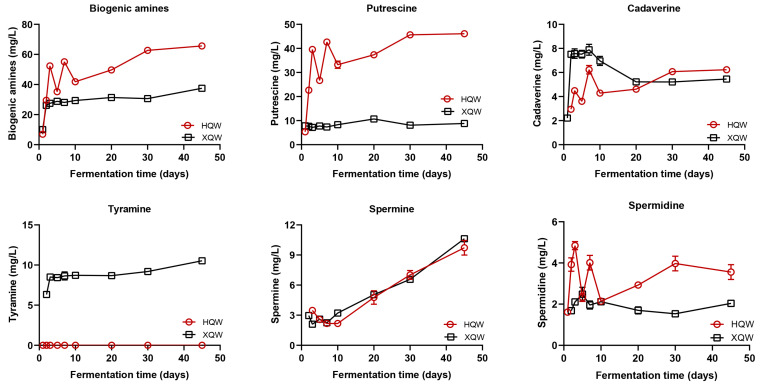
The dynamic change in biogenic amines (BAs, including putrescine, cadaverine, tyramine, spermine, and spermidine) during the brewing of HQW and XQW.

**Figure 4 foods-13-02452-f004:**
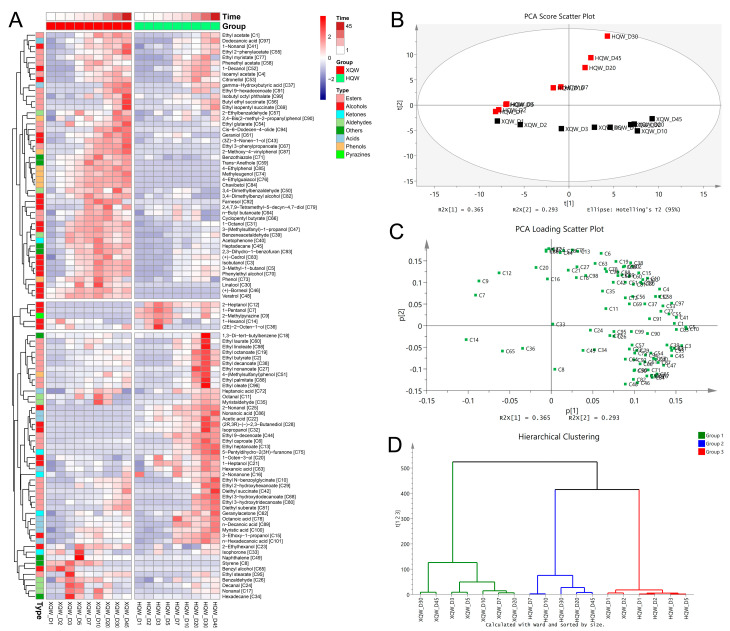
The dynamic changes in the volatile flavor components during the brewing of HQW and XQW. (**A**) A heatmap of the abundances of volatile components. (**B**) A principal component analysis (PCA) score scatter plot. (**C**) A PCA loading scatter plot. (**D**) A hierarchical clustering diagram.

**Figure 5 foods-13-02452-f005:**
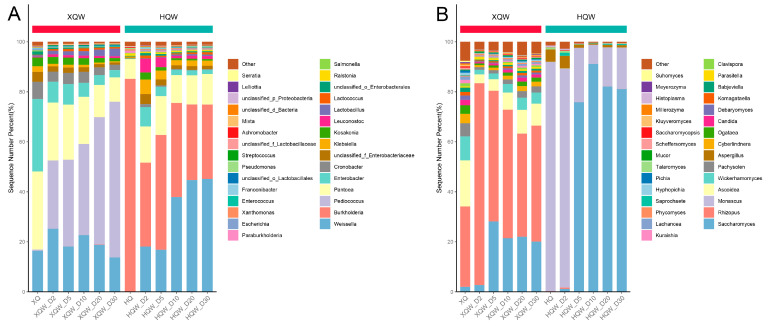
A stacked histogram of the relative abundance of the predominant bacterial (**A**) and fungal (**B**) genera during the HQW and XQW brewing processes.

**Figure 6 foods-13-02452-f006:**
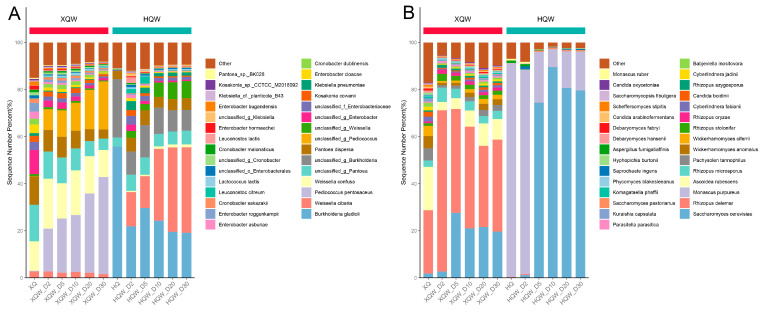
A stacked histogram of the relative abundance of the predominant bacterial (**A**) and fungal (**B**) species during the HQW and XQW brewing processes.

**Figure 7 foods-13-02452-f007:**
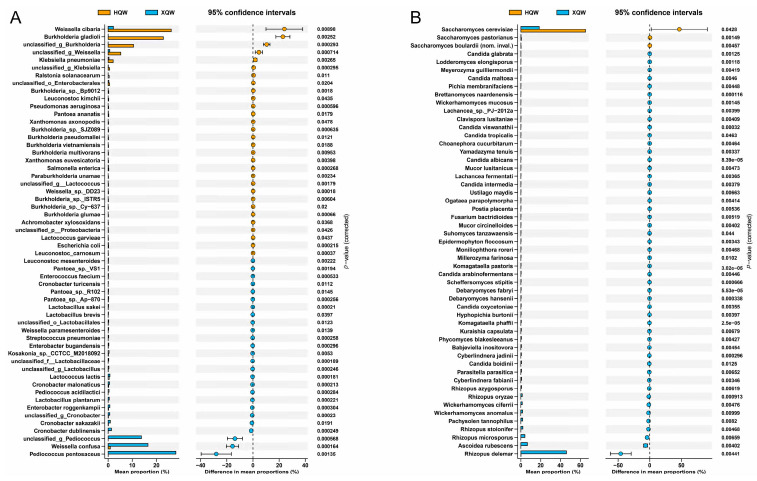
A visualization of the differences in the relative abundance of bacterial (**A**) and fungal (**B**) species between HQW and XQW. Microbial species with significant differences between HQW and XQW were determined using Welsh’s *t*-test, and the Benjamini–Hochberg procedure was used to control the false discovery rate due to multiple tests. The corrected *p* values are shown on the right.

**Figure 8 foods-13-02452-f008:**
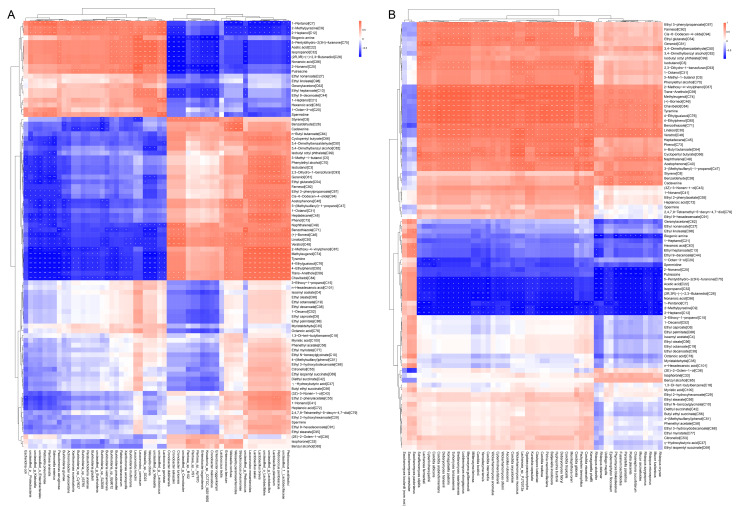
A correlation analysis between the characteristic volatile components, biogenic amines (BAs), and the predominant microbial phylotypes at the species level during the HQW and XQW brewing processes. (**A**) Key bacterial species—volatile components/biogenic amines (BAs); (**B**) key fungal species—volatile components/biogenic amines (BAs).

**Figure 9 foods-13-02452-f009:**
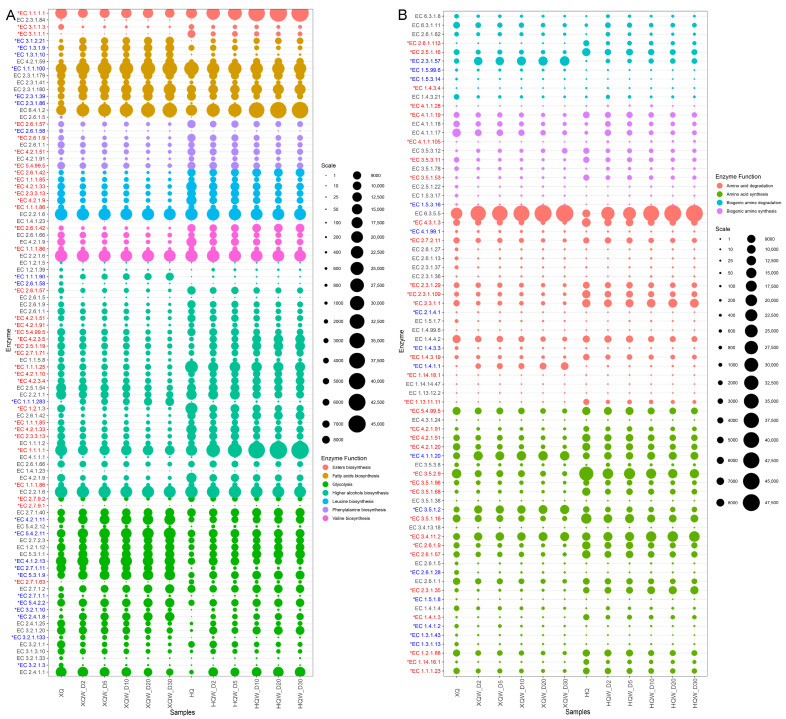
A bubble chart of the abundances of microbial genes for enzymes closely related to the metabolism of characteristic volatile components (**A**) and biogenic amines (**B**) during the HQW and XQW brewing processes.

## Data Availability

The original contributions presented in this study are included in the article/Appendix A; further inquiries can be directed to the corresponding author.

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
