# Peer review of "A Comparative Study of Microbial Communities, Biogenic Amines, and Volatile Profiles in the Brewing Process of Rice Wines with Hongqu and Xiaoqu as Fermentation Starters"

_foods, 2024, doi:10.3390/foods13152452_

Round 1

Reviewer 1 Report

Comments and Suggestions for Authors

The manuscript written by Yan et al., compares the biogenic amine content (BA), the microbial population and the volatile compounds in two different rice wines.

In my opinion, the work is complete, there is a lot of data both microbiological and chemical.

I appreciate the unique section results and discussion instead of separate sections because due to the large amount of data, this helps to better follow the different analyses made.

I observed some aspects to be reviewed.

Paragraph 3.1.

It is not possible to produce 100g/l of ethanol from 50g/l of sugars, please check the data

How do you explain that HQW, which contains more sugar, at the end of alcoholic fermentation has a minor ethanol concentration? Comment on this data.

Regarding the acidity, what are the common values found in these beverages? Are there any limits?

Line 239: I didn’t find the method for quantifying amino nitrogen.

Paragraph 3.3.

This part should be deeply discussed. Is there a regulatory limit concerning the concentration of biogenic amines in wine?

Did you quantify the amount of precursor of BA in the different wines?

Paragraph 3.4.

You quantified the VOCs during the fermentation. Are the average values reported in table 1?

Lines 328-331: since that the samples XQW_D1-D2 and HQW_D1-D5 are in the same group, can be affirmed that in the first two days of fermentation, the VOCs are similar, and they differ during fermentation?

The discussion is missing, you just exposed the results.

Paragraph 3.6.

The discussion is missing, you just exposed the results.

A final comment: due to the high BA content, is it possible to change the composition of these Qu to reduce BA production during fermentation?

Thank you for this opportunity.

Best regards

Reviewer 2 Report

Comments and Suggestions for Authors

Comments on the Quality of English Language

Minor editing of English language is required.

Author Response

Reviewer #2:

The manuscript is interesting and well-written. It contributes new knowledge to the study of the microbial communities contributing to the fermentation of rice wines and their effect on flavor quality. The research results are important for the field of production and quality of rice wines. The manuscript cites relevant literature. A few comments to improve the manuscript can be found below.

Q1: The abstract is extended; however, it doesn’t report the methodology used for the experiment.

R1: Thanks for your valuable suggestion. We have revised the Abstract section, as detailed in the revised resubmission.

Abstract: “...... In this study, GCMS, GC, HPLC and metagenomic sequencing techniques were used to compare the microbial communities, biogenic amines and volatile profiles during the brewing of rice wines with Hongqu and Xiaoqu as fermentation starters.”

Q2: The objectives of the study are clearly stated in this section. The authors should elaborate on the research gap and novelty of this study in the Introduction secttion.

R2:Thanks for your valuable suggestion. We have revised the Introduction section, as detailed in the revised resubmission.

Introduction: “......Certain researchers employed high-throughput sequencing (HTS) methodology to explore disparities in microbial communities and volatile metabolites across various traditional starter cultures[8]. However, microbiological analysis is often confined to the genus level, potentially rendering the outcomes of such studies incongruent with actual conditions. Metagenomics offers a valuable perspective by elucidating the biological relevance of microbial species that harbor crucial genes in the brewing environment.”

References:

[8] Huang, Z.-R.; Guo, W.-L.; Zhou, W.-B.; Li, L.; Xu, J.-X.; Hong, J.-L.; Liu, H.-P.; Zeng, F.; Bai, W.-D.; Liu, B.; et al. Microbial communities and volatile metabolites in different traditional fermentation starters used for Hong Qu glutinous rice wine. Food Research International 2019, 121, 593-603.

Q3: A graphical abstract including the production process of rice wines would be beneficial for readers to better understand the overall process.

R3: Thanks for the reviewer's valuable comments. We have submitted a graphical abstract in the manuscript accordingly. Please refer to the revised version we have submitted for details.

Q4: Line 65. Please correct compared to “compare”.

R4: Thanks for your valuable advice. We have revised the manuscript in accordance with your suggestions. For details, please refer to the revised manuscript we submitted.

Q5: Line 75. What is considered an appropriate amount? Please give an indication.

R5: Thanks for your valuable advice. We have revised the manuscript in accordance with your suggestions. For details, please refer to the revised manuscript we submitted.

Abstract: “......It was found that appropriate amount of high alcohols (<300mg/L) ...... ”

Q6: Lines 77-78. What is considered a high level? Please give an indication.

R6:Thanks for your valuable advice. We have revised the manuscript in accordance with your suggestions. For details, please refer to the revised manuscript we submitted.

Abstract: “......However, high level of higher alcohols (>400mg/L) ......”

Q7: Line 120. Please report the material of the plasstic film.

R7:Thanks for your valuable advice. We have revised the manuscript in accordance with your suggestions. For details, please refer to the revised manuscript we submitted.

Section 2.1:“......From the tenth day of fermentation, 8 layers of gauze were substituted with glass bottle gap to facilitate anaerobic fermentation (18℃ for 30 days).”

Q8: Lines 121-122. How was the sampling done?

R8: Thanks for your valuable advice. We have revised the manuscript in accordance with your suggestions. For details, please refer to the revised manuscript we submitted.

Section 2.1:“......First, a precise 10 gram sample was weighed and added to it 90 mL of 0.85% normal saline.”

Q6: Line 125. Parameter should be corrected to “parameters”

R6: Thanks for your valuable suggestion, we have corrected it in revised manuscript.

Q7: 2.2. Physiochemical parameter determination. Please correct to Physicochemical

R7: Thanks for your valuable advice. We have revised the manuscript in accordance with your suggestions.

Q8: 2.4. Biogenic amines determination Lines 148-150. Please revise these lines so they don’t sound like reading instructions.

R8: Thanks for your valuable suggestions. According to your comments, we have made improvements to the manuscript.

Section 2.1:“......The biogenic amine content in the fermentation samples was determined by high performance liquid chromatography (HPLC) which based on the Chinese standard GB 5009.208-2016.”

Q9: Line 222. Please delete the word “trend”.

R9: Thanks for your valuable advice. We have revised the manuscript in accordance with your suggestions.

Q10: Line 239. There is a description of amino nitrogen content, but the analysis is not mentioned in the Materials and Methods section.

R10: Thanks for your valuable suggestions. According to your suggestions, we have made corresponding supplements in the Materials and Methods section of the revised manuscript.

Section 2.2: “......Amino nitrogen content was measured according to the Chinese official method (GB/T 13662-2018 Rice Wine). ”

Q11: Lines 239-240. Why did the nitrogen content increase during fermentation? Please, elaborate.

R11:Thanks for your valuable suggestions. According to your suggestions, we have made corresponding supplements in the Materials and Methods section of the revised manuscript.

Section 3.1: “......As a key component, amino acids were mainly derived from the hydrolysis of proteins in raw materials. Taking rice wine brewing as an example, glutinous rice, as one of the commonly used raw materials, was rich in protein. During the brewing process of rice wine, protease and carboxypeptidase secreted by Aspergillus and monascus act on proteins in glutinous rice to produce amino acids or oligopeptides. This may be the reason for the increase of amino acid nitrogen during rice wine brewing. ”

Q12: Table 1. Please change the font to Palatino Linotype as per the journal instructions.

R12: Thanks for your valuable suggestions. We have changed the font of Table 1 to Palatino Linotype.

Q13: References are not appropriately cited from what I could assess (abbreviated journal name, year in bold, volume in italics). The rate of self-citation is 20,6%, exceeds the journal’s acceptable range.

R14:Thanks for your valuable advice. We have revised the manuscript in accordance with your suggestions.

Reviewer 3 Report

Comments and Suggestions for Authors

The manuscript evaluated the influence of the fermentation starter on rice wine characteristics. The work is in the scope of the journal, the ways and means are clearly described and well organized. However, there are points that need to be clarified:

·       Abstract is too long with too descriptive results. In addition, the methodology is missing. An abstract must contain an introduction, objective, methodology, main results and discussion. Please revise it.

·       Wheatqu is not explained in introduction. Please add it.

·       What is the novelty of the work? There are many works from the same authors exploring the same subject (https://doi.org/10.1021/acsomega.3c04732, https://doi.org/10.1016/j.foodres.2023.113488 ), specially in concern of HQW.

·       Line 117: why to use different amount of Hongqu (10%) and Xiaoqu (1%)? This can have an impact on saccharification step, which will be faster in HQW due to the presence of 10 times more fermenter starter.

·       Item 2.1: What was the volume used in rice wine production?

·       Line 181: what is jiupei?

·       Line 232: when authors say “HQW at the end of fermentation was significantly lower in comparison to XQW”, which statistical analysis was performed to ensure the significance?

·       Line 240: This sentence is confusing “and no significant difference was observed in amino nitrogen content between HQW and XQW brewing, but the content of amino nitrogen in HQW was higher than that in XQW at the end of brewing.”

·       Please avoid describing figures in the text, this is repeated all over the manuscript. For example, in item 3.2 authors describe figure 2 but the comparison with literature is missing. Item 3.5 the same happens.

·       Line 277: spermidine is repeated

·       Figure 4a is difficulty to read. Please enlarge it or move to a supplementary material.

·       Figures 5, 6, 7, 8 and 9 resolution is not good. As it is, it is not possible to read the data.

·       Item 3.6 : data from figure 8 is only described in the text without discussion in a too long paragraph.

·       Lines 503-565: this is a too long paragraph

·       In general, results must be discussed and compared with literature.

·       The conclusion should not be a summary of the work. In fact, there is a lot of repeated sentences in conclusion and abstract. Please correct the conclusion section.

Author Response

Response to Editor & Reviewers

Journal: Foods

Manuscript ID: foods-3001236

Manuscript Title: Comparative study of microbial communities, biogenic amines and volatile profiles in the brewing process of rice wines with Hongqu and Xiaoqu as fermentation starters

Reviewer #3:

The manuscript evaluated the influence of the fermentation starter on rice wine characteristics. The work is in the scope of the journal, the ways and means are clearly described and well organized. However, there are points that need to be clarified:

Q1: Abstract is too long with too descriptive results. In addition, the methodology is missing. An abstract must contain an introduction, objective, methodology, main results and discussion. Please revise it.

R1: Thanks for your valuable suggestion. We have revised the Abstract section, as detailed in the revised resubmission.

Abstract: “......In this study, GCMS, GC, HPLC and metagenomic sequencing techniques were used to compare the microbial communities, bioamines and volatile components of rice wine with Hongqu and Xiaoqu as fermentation starter.”

Q2: Wheatqu is not explained in introduction. Please add it.

R2: Thanks for your valuable suggestions. According to your comments, we have made corresponding additions and improvements to the manuscript, and your comments have greatly helped to improve the quality of our manuscript. For details, please refer to the revised manuscript we submitted.

Abstract: “......Wheat Qu was made from raw wheat, which naturally contains mould, bacteria and yeast. In the production process, the wheat was usually crushed, mixed with water, and pressed into brick-like blocks by manual trampling or mechanical force[66] .”

References:

[66] Zhang, K.; Li, Q.; Wu, W.; Yang, J.; Zou, W. Wheat Qu and Its Production Technology, Microbiota, Flavor, and Metabolites. Journal of Food Science 2019, 84, 2373-2386.

Q3: What is the novelty of the work? There are many works from the same authors exploring the same subject (https://doi.org/10.1021/acsomega.3c04732, https://doi.org/10.1016/j.foodres.2023.113488 ), specially in concern of HQW.

R3:Thanks for your valuable suggestion. We have revised the Introduction section, as detailed in the revised resubmission.

Introduction: “......Certain researchers employed high-throughput sequencing (HTS) methodology to explore disparities in microbial communities and volatile metabolites across various traditional starter cultures[8]. However, microbiological analysis is often confined to the genus level, potentially rendering the outcomes of such studies incongruent with actual conditions. Metagenomics offers a valuable perspective by elucidating the biological relevance of microbial species that harbor crucial genes in the brewing environment. In this study, we contrasted the temporal variations of the dynamics of biogenic amines, higher alcohols and volatile profiles in the brewing process of rice wines fermented with Hongqu and Xiaoqu (HQW and XQW). ”

References:

[8] Huang, Z.-R.; Guo, W.-L.; Zhou, W.-B.; Li, L.; Xu, J.-X.; Hong, J.-L.; Liu, H.-P.; Zeng, F.; Bai, W.-D.; Liu, B.; et al. Microbial communities and volatile metabolites in different traditional fermentation starters used for Hong Qu glutinous rice wine. Food Research International 2019, 121, 593-603.

Q4: Line 117: why to use different amount of Hongqu (10%) and Xiaoqu (1%)? This can have an impact on saccharification step, which will be faster in HQW due to the presence of 10 times more fermenter starter.

R4:Thanks for your valuable advice. We are very sorry that we did not describe this clearly .

This is due to the different requirements of the HQW and XQW brewing processes.

Q5: Item 2.1: What was the volume used in rice wine production?

R5:Thanks for your valuable advice. Since this study simulated traditional brewing in the laboratory. We used 1L fermenters to brew Chinese rice wine, and the amount of brewing was determined by the experimental arrangement.

Q6: Line 181: What is jiupei?

R6: We apologize for not mentioning the exact meaning or definition of "jiupei", which would have caused great inconvenience to the readers in reading the article as they could not quickly understand the content of the article. Following your suggestion, we have clarified the definitions of jiupei in the revised manuscript. The specific content of the modification is as follows:

Section 2.6“......Jiupei referred to the unfiltered wine during the brewing process”

Q7: Line 232: when authors say “HQW at the end of fermentation was significantly lower in comparison to XQW”, which statistical analysis was performed to ensure the significance?

R7: Thanks for your valuable advice. We are very sorry that we did not describe this clearly in our previous submission, and in fact we did statistically analyze the experimental results. Statistical analysis of the data was performed using Statistical Package for the Social Sciences (SPSS) 23.0 (IBM, Armonk, New York). Student’s t-test was used for two-group comparisons (*p < 0.05, **p < 0.01).

Q8: Line 240: This sentence is confusing “and no significant difference was observed in amino nitrogen content between HQW and XQW brewing, but the content of amino nitrogen in HQW was higher than that in XQW at the end of brewing.”

R8: Thanks for your valuable advice. We are very sorry that we did not describe this clearly .We have adjusted and modified this section according to your proposal.

Introduction: “...... In this study, the content of amino nitrogen during HQW and XQW brewing increased with fermentation, and the content of amino nitrogen in HQW was higher than that in XQW at the end of brewing.”

Q9: Please avoid describing figures in the text, this is repeated all over the manuscript. For example, in item 3.2 authors describe figure 2 but the comparison with literature is missing. Item 3.5 the same happens.

R9: Thanks for your comments.We are very sorry that we did not describe this clearly .We have adjusted and modified 3.2 and 3.5 section according to your proposal.

Q10: Line 277: spermidine is repeated

R10: Thanks for your valuable suggestions. We have removed the duplicates as per your suggestion.

Q11: Figure 4a is difficulty to read. Please enlarge it or move to a supplementary material.

R11:Thank you for your valuable suggestions. We apologize for the inconvenience of low-resolution images. Therefore, we have revised the image to a higher resolution. For details, please refer to the revised manuscript we submitted.

Q12: Figures 5, 6, 7, 8 and 9 resolution is not good. As it is, it is not possible to read the data.

R12: Thank you for your valuable suggestions. We apologize for the inconvenience of low-resolution images. Therefore, we have revised the image to a higher resolution. For details, please refer to the revised manuscript we submitted.

Q13: Item 3.6 : data from figure 8 is only described in the text without discussion in a too long paragraph.

R13: Thanks for your valuable advice. We have adjusted and modified this section according to your proposal.

Q14: Lines 503-565: this is a too long paragraph

R14: Thanks for your comments. We have refined the paragraph to make it more concise

Q15: In general, results must be discussed and compared with literature.The conclusion should not be a summary of the work. In fact, there is a lot of repeated sentences in conclusion and abstract. Please correct the conclusion section.

R15: Thanks for your valuable advice. We have adjusted and modified this section according to your proposal.

Section 4“......In this research, we compared the microbial ecology, biogenic amine content, and volatile compounds during the HQW and XQW brewing. Our findings indicated that XQW contained more higher alcohols compared to HQW. Conversely, HQW exhibited a significantly higher content of BAs. The volatile flavor components of both HQW and XQW exhibited distinct profiles. Metagenomic sequencing further revealed that the brewing processes of HQW and XQW were governed by distinct microbial populations. Moreover, we identified microbial genes responsible for the metabolism of biogenic amines and higher alcohols in both wines. Notably, HQW possessed a higher abundance of genes related to BAs synthesis and amino acid metabolism compared to XQW. Conversely, XQW exhibited a significantly higher abundance of genes linked to fatty acid metabolism. This comprehensive analysis provides crucial scientific insights into how microbial communities influence the flavor quality of HQW and XQW, paving the way for selecting beneficial microorganisms to enhance the flavor profiles of rice wine. Therefore, the subsequent study can employ multi-omics integration technology for a more comprehensive and systematic analysis of the metabolic mechanism of BAs and higher alcohols.”
